# Influence of Dietary Lipid Source Supplementation on Milk and Fresh Cheese from Murciano-Granadina Goats

**DOI:** 10.3390/ani13233652

**Published:** 2023-11-25

**Authors:** Francisco Moya, Josefa Madrid, Fuensanta Hernández, Irene Peñaranda, María Dolores Garrido, María Belén López

**Affiliations:** 1KPRA Leche y Genética, S.L., Autovía del Noroeste (C-415) Salida n° 13 Dirección Fuente Librilla, Km—4.0 Paraje «La Alquibla», 30170 Mula, Spain; 2Department of Animal Production, Faculty of Veterinary Science, International Excellence Campus for Higher Education and Research “Campus Mare Nostrum”, University of Murcia, Campus de Espinardo, 30100 Murcia, Spain; alimen@um.es (J.M.); nutri@um.es (F.H.); 3Department of Food Science and Technology, Faculty of Veterinary Science, International Excellence Campus for Higher Education and Research “Campus Mare Nostrum”, University of Murcia, Campus de Espinardo, 30100 Murcia, Spain; irene.penaranda@um.es (I.P.); mgarrido@um.es (M.D.G.)

**Keywords:** goat, flaked linseed, salmon oil, fatty acid, sensory profile

## Abstract

**Simple Summary:**

Several studies have studied the influence of diet supplementation on goat milk quality, but the number of animals included is normally small or does not reflect real commercial conditions. Moreover, little technological and sensory input is provided. Thus, our study included 350 goats divided into three groups (one control, two supplemented with ingredients rich in n-3). Dietary n-3 treatments modify the fatty acid profile without making any sensory difference to milk and fresh cheese, accompanied by marginal modifications to the physicochemical profile. Therefore, milk obtained from animals receiving dietary supplementation can be provided to the dairy industry.

**Abstract:**

This study analyzes the influence of the incorporation of flaked linseed and fish oil in the diet on the resulting milk and cheese. Three dietary treatments were assayed in 350 milking Murciano-Granadina multiparous goats in full-lactation: a control diet and two experimental diets, one including flaked linseed (FL) at 3.88% of dry matter, and the other containing salmon oil (SO) at 2.64% of dry matter for three periods of 21 d. None of the dietary treatments affected the daily milk yield, cheese yield, or the physicochemical parameters of the milk and cheese. Regarding the fatty acid profile (FA), the milk and cheese from animals whose diets were supplemented with SO had a higher percentage of fatty acids than those obtained with the FL-supplemented diet, except for C18:0, C18:1, C18:2 n-6, trans-9, trans-12 C18:2, cis-9, trans-11 C18:2, C18:3, and C19:0, which reached their highest levels in milk obtained with the diet supplemented with FL. The decrease in the percentage of C16:0 was greater in the milk derived from the FL diet than from the SO diet. The FL-supplemented diet improved the nutritional value of milk due to a reduction in saturated fatty acids (SFAs) and increases in polyunsaturated fatty acids (PUFAs) and conjugated linoleic acid (CLA). The decrease in n-6/n-3 in the observed milk was more pronounced with the FL diet. No differences in the sensory profile were found for the milk and cheese derived from the different dietary treatments. Dietary n-3 treatments modified the fatty acid profile without making any sensory difference to milk and fresh cheese, accompanied by marginal modifications to the physicochemical profile. We conclude that dietary supplementation with flaked linseed or fish oil produces milk and cheese from Murciano-Granadina goats with a higher nutritional quality without modifying the sensory profile of the corresponding products obtained from animals that were fed a routine diet.

## 1. Introduction

Goat milk is the third most produced milk variety in the world, with an increase of more than twofold in recent decades, and an expected increased market probability of 53% by 2030 [1]. There are several differences in its composition compared with cow’s milk that determine their low allergenic potential, digestibility, and nutritional value, and contribute that goat milk can be considered a natural functional food that its consumption should be promoted [2].

Murciano-Granadina goat milk is characterized by an excellent technological aptitude to produce different cheese varieties, where cheese quality is determined by fat and protein composition [3]. Modification of the basal diet, especially with dietary fat sources in ruminant diets, could be a good strategy to reduce the levels of saturated fatty acids (SFAs) in accordance with public health policies [4] that recommend a reduction in the consumption of saturated fatty acids (SFAs) and an increases in the consumption of n-3 FAs, especially α-linoleic acid (ALA, C18:3n-3), eicosapentaenoic acid (EPA, C20:5n-3), and docosahexaenoic acid (DHA, C22:n6-3), which have health benefits. Grasses and vegetable seeds or oils such as linseed, hemp seed, chia seed, and rubber seed are sources of α-linoleic acid, while fishmeal and oil, algal oil, and microalgae biomass are sources of eicosapentaenoic acid and docosahexaenoic acid [5]

Most studies have been performed in dairy cows; however, it has been suggested that there are differences in lipid biohydrogenation pathways in the rumen, and in lipogenesis in the mammary gland between cows and goats [6]. Thus, it has been found that among 19 mRNAs encoded by genes related to lipid metabolism in the mammary gland, 15 of them exhibited a different abundance, where 9 were more abundant in cows and 6 were more abundant in goats [7]. Furthermore, Fougère et al. (2018) claimed that various lipid sources could have different effects on the lipid profile or fat content of milk, and the profile is also dependent of the ruminant species (goats or cows) [8].

Additionally, one problem observed with lipid supplementation in dairy cows is the decrease in protein content, which modifies the coagulation properties and may alter the texture of the final product [9].

Although most studies agree that the final concentrations of PUFAs and CLA in processed dairy products are mainly associated with their content in milk, the fatty acid profile of milk and dairy products, especially the CLA content, may also be associated with the effects of technological processes applied to obtain products such as cheese [10].

Nowadays, it is technically feasible to modify FAs through feeding strategies; however, the effects of such supplementation on the properties of milk and cheese produced from the Murciano-Granadina goat breed are scarce and need to be further investigated and constitute the proposal of this study.

Several articles have studied the influence of diet supplementation on goat milk quality [7,8,9]; however, the numbers of animals included are normally small or do not reflect real commercial conditions. Moreover, little technological and sensory input is provided.

The main aim of this study was to improve the nutritional value of the FAs of goat milk and fresh cheese by supplementing the diet of Murciano-Granadina goats with flaked linseed (FL) and salmon oil (SO) as lipid sources. This study represents a preliminary stage to validate the use of these supplementations at an industrial level in the production of matured cheeses.

## 2. Materials and Methods

### 2.1. Animals and Treatments

The experimental procedures used in this study followed the Directive 2010/63/EU and Royal Decree 53/2013, which laid down basic rules for the care and handling of research animals. The experiment was conducted in a farm under an intensive production system located in the southeast of Spain (Kpra Leche y Genética, S.L., Mula, Murcia, Spain). In total, 350 milking Murciano-Granadina multiparous goats (average body weight of 44.65 ± 3.48 kg) in full lactation were distributed into three groups of 150, 100, and 100 animals according to parity and their daily milk yield (recorded one week before the trial). The study followed a 3 × 3 crossover design, with 3 periods of 21 d each (14 d for adaptation and 7 d for sampling and data recording).

In period 1, each group was randomly assigned to 1 of the 3 dietary treatments, rotating the treatments in period 2 and 3, until all had received all of the experimental treatments. The three dietary treatments were a control diet, supplemented with calcium soaps of palm oil FA an ingredient with 84% ether extract (MAGNAPAC^®^, Norel Animal Nutrition, Madrid, Spain) and two experimental diets, supplemented with ingredients rich in n-3—one that included flaked linseed (FL) with 34.7% ether extract (broken and hydrothermally processed from Agrocava S.L. company, Caravaca de la Cruz, Murcia, Spain), an ingredient rich in ALA—and another that contained salmon oil (SO) concentrate, a product with 50% ether extract (Optomega-50, Optivite International Ltd., Nottinghamshire, UK), a supplement rich in DHA and EPA. Each diet was in the form of a total mixed ration (TMR) consisting of alfalfa hay, citrus pulp, a concentrate mixture, and a lipid-rich supplement partially substituting the concentrate (1.55%, 3.88%, or 2.64% of diet on a dry basis) with calcium soaps of palm oil FAs, FL, or SO concentrate, respectively. All diets adhered to the nutritional requirements for lactating goats according to the Fundación Española para el Desarrollo de la Nutrición Animal [11] and they were formulated to be iso-energetic and iso-nitrogenous. The ingredients and chemical composition of the diets are shown in Table 1. The animals were fed the TMR ad libitum twice daily (at 08:00 and 16:00 h) with sufficient feed for approximately 5% to remain uneaten. The amount of feed offered and refused was recorded daily throughout the experimental period. Goats had unlimited access to water and were milked twice a day. Milk yield per group and day was recorded daily during the sampling period, and samples of the offered and refused diet of each period and group were collected to determine the dry matter (DM) content and for chemical analysis.

#### Feed Analyses

The DM contents of the feed and orts were determined by oven-drying at 60 °C for 48 h. Dried feed samples were analyzed for crude protein (CP, total nitrogen × 6.25) and ether extract (EE [12] for starch (Ewers’ polarimetric method) [13]), and for neutral detergent fiber (NDF), acid detergent fiber (ADF), and acid detergent lignin (ADL), using the procedure developed by Van Soest et al. [14] with filter bag equipment obtained from Ankom (Ankom Technology Corp., Fairport, NY, USA). In addition, TMRs were analyzed to determine their fatty acid composition. Fatty acid extractions in the feed samples were performed according to the method developed by Folch et al. [15] and methylation to finally determine the fatty acid profiles of the milk and cheese using the gas chromatography technique.

### 2.2. Fresh Goat-Cheese-Making

Bulk milk samples were collected from each group during the sampling period. On two days of each period, 15 L samples of milk were used for chemical and sensory analysis and for cheese-making. The milk was pasteurized (78 °C for 15 s) using a plate heat exchanger (100 L Alfa Laval, Lund, Sweden) in the pilot plant of the Food Technology Department on the same day of milking, and immediately stored at 4 °C. Cheese-making was carried out the following day according to Garcia et al. [16] Each type of pasteurized Murciano-Granadina goat milk was introduced in a double-zero cheese-vat (Type 10 L, Pierre Guerin Technologies, Mauze, France) and tempered until a constant temperature of 32 °C. Stirring slowly, 3 mL of CaCl_2_ (Chr. Hansen, Horsholm, Denmark) at a concentration of 510 g/L was added. Then, 3 mL of calf liquid rennet (Caglio Star Spain S.A., Cieza, Murcia, Spain) was added and the cutting time was determined by multiplying T_max_ by β = 3, as a modification of the method described by Fagan et al. [17]. Tmax is useful for predicting milk clotting time; it is an optical parameter derived from a CoAguLiteTM optical sensor (Reflectronics, Inc., Lexington, KY, USA), which is coupled to the vat.

When the milk clotting time was reached, the curd was cut in a first cut of 20 s, followed by a pitching of 10 min, and then a second cut of 5 min. Another pitching of 3 min was performed before finally stirring for 15 min. The curds were placed in square molds, unpressed, and brined (17 °Be for 30–40 min). Cheese yield was defined as the amount of milk needed to obtain a certain weight of cheese (L kg^−1^).

### 2.3. Physicochemical Analysis of Milk and Cheese

All the analyses were performed in triplicate. The milk pH measurements were taken with a pH meter (Crison^®^ micro pH 2001, Barcelona, Spain) connected to a pre-calibrated Crison^®^ glass combined electrode (1952–2002). The pH of the cheeses was measured by suspending 5 g of grated cheese in 30 mL of distilled water and stirring for 10 min. DM was determined according to IDF [18]. The fat content of the milk and cheese was measured using the Van Gulik method [19]. The milk and cheese protein contents were determined using the Kjeldahl method [20]. To obtain the FA contents of each type of milk and cheese, lipids were extracted according to the Röse-Gottlieb method before derivatization and quantified by gas chromatography [21] using a Finnigan Trace GC ULTRA gas chromatograph equipped with an AS3000 auto-sampler (both from ThermoFinnigan, Madrid, Spain), a capillary column with a 70% cross-linked CyanopropylPolysilphenylene-siloxane, a 60 m length, a 0.25 mm internal diameter, and a 0.25 μm film thickness (BPX70, Kinesis Australia Pty Ltd., Redland Bay, Australia), and an ionization flame detector. The methyl esters of FA were identified by comparison with the retention times of reference standards (Sigma-Aldrich, St. Louis, MO, USA). The FA integration was processed via software from Chrom Card Fisons Instruments (https://photos.labwrench.com/equipmentManuals/12480-4887.pdf (accessed on 3 August 2023) Fison Instruments, Glasgow, UK), and the fatty acid composition was expressed on a quantitative basis in milligrams of fatty acid per 100 g serving of milk or cheese, based on the fat content (g/100 g) of each type of product. The injections were performed in triplicate. The atherogenic (AI) and thrombogenic indices (TIs) were calculated according to Ulbricht and Southgate (1991). AI = (12:0 + 4 × 14:0 + 16:0)/(MUFA + [n-3 + n-6]); TI = (14:0 + 16:0 + 18:0)/(0.5 × MUFA + 0.5 × n-3 + 3 × n-6 + n-3/n-6).

### 2.4. Texture and Sensory Analysis

Cheese texture profile analysis (TPA) was determined according to García et al. [12]. TPA was carried out using a texture analyzer QTS-25 (Brookfield CNS Farnell, Borehamwood, UK) equipped with a loadcell of 25 kg and Texture Pro V. 2.1 software. The conditions were as follows: 20 °C room temperature; two consecutive cycles of 50% compression; crosshead at a constant speed of 30 mm.min^−1^; and a trigger point of 0.05 N. The texture parameters, hardness 1 (N), hardness 2 (N), gumminess (N), chewiness (N*mm), cohesiveness (adimensional), adhesiveness (N/mm^2^), and springiness (mm), were analyzed.

Sensory profiles of milk and cheese were analyzed by 8 trained panelists according to García et al. [12], chosen from the staff of the Department of Food Technology of the University of Murcia; all of them had experience in the evaluation of profiles of different dairy products.

A prescribed descriptive test [22] was carried out after previous training of the panelists, when every parameter was defined and quantified. Granny Smith apples, crackers, and natural mineral water were served to cleanse the palate between samples.

Sensory milk and cheese parameters were assessed by the 8 panelists in a laboratory equipped for sensory analysis, at the Food Technology Department of the University of Murcia. Refrigerated fluid milk was tempered at 16 °C and the samples (approximately 35 mL) were presented to the testers in plastic cups with random code numbers. Sensory analysis of the cheeses was performed 24 h after the cheese-making process and each cheese was split into two halves and labeled using randomly chosen digits. The order of sample presentation within each session was balanced to account for order and carryover effects [23]. One was divided into wedges approximately 1 cm thick. The other half was used for the visual phase.

This score set included eight descriptors for milk associated with odor and flavor, and ten descriptors for cheese associated with odor, flavor, and texture, all of them with a structured intensity scale ranging between 1 and 10. Both milk and cheese tests included a global quality score in the same range.

### 2.5. Statistical Analysis of the Results

Statistical treatment of the physicochemical data was performed using IBM SPSS Statistics 19 (IBM Spain, S.A., Madrid, Spain). The data were analyzed with a repeated measures linear mixed model, with diet as the fixed effect and group and phase as random factors. Pairwise comparisons among means were performed using the least significant difference (LSD). The level of significance was taken as *p* < 0.05, with a trend towards significance at *p* < 0.1. Statistical treatment of the sensory data was performed with Minitabv 15.0 (Addlink Software Scientific, S.L. Barcelona, Spain). One-way ANOVA was used to determine significant differences.

## 3. Results

### 3.1. Goat Milk and Cheese Composition

DM intake (Table 2) tended to be affected by dietary treatment (*p* < 0.1); SO intake was lower than with the control diet; and FL intake did not significantly differ from the other diets. The dietary treatments had no significant effect on daily milk yield. As shown in Table 2, in general, the supplementation of goat diets had no effect on the physicochemical parameters of the resulting milk, since no significant differences (*p* > 0.05) were observed in pH, protein, or fat. However, significant differences were observed in the dry matter content of the milk (*p* < 0.05), which was lower in the milk from animals receiving the SO diet than in the corresponding milk from the FL diet.

The type of supplementation did not influence the pH, DM, protein, or fat content of the fresh cheese (*p* > 0.05) (Table 3). Although slight decreases in fat were observed in cheeses derived from animals fed the SO-supplemented diet, no significant differences (*p* > 0.05) were observed in the clotting time (T_max_) or cheese yield. A higher T_max_ and lower yield were determined in cheeses derived from animals fed the FL-supplemented diet.

### 3.2. Fatty Acid Profile

Regarding the milk FA content (Table 4), significant differences (*p* < 0.01) among treatments were found for all FAs, except C4:0. The milk from goats that were fed the SO-supplemented diet exhibited higher percentages of most FAs than the milk from goats fed the control diet, although C16:0, C18:0, and C18:1 had lower values than the control. Notably, the SO diet was rich in DHA and EPA; therefore, an increase in the fatty acids in the milk was to be expected. Comparing the FA values of milk obtained from the SO and FL diets, higher percentages were found with the SO diet, except for C18:0, C18:1, C18:2, trans-9, trans-12 C18:2, cis-9, trans-11 C18:2; C18:3, and C19:0, which reached their highest levels in milk derived from goats fed the FL diet. The 249% increase in concentration of ALA in the diet supplemented with FL was expected due to the high concentrations of the FA of this type of seed. In contrast, a lower percentage of palmitic acid (C16:0) was obtained in the milk derived from animals fed with FL than with the SO-supplemented diet.

Rumenic acid (RA, C18:2, cis-9, trans-11) levels were highest in the diet supplemented with FL. Non-conjugated isomers of C18:2 were detected, and the percentages of C18:2, trans-9, and trans-12 were higher in the milk of animals receiving FL/SO diets. One of the highest PUFA concentrations in our study was found to be arachidonic acid (C20:4 n-6). In the milk obtained from the goats that were given FL, the concentrations of SFAs (−1.88%) were lower, while those of PUFAs (36.03%), CLAs (48.81%), n-3 (197.59%), and n-6 (11.87%) were higher than in the control milk. The n-6/n-3 ratio was the lowest (−62.58%). Goat milk obtained with the diet supplemented with SO showed higher concentrations of SFAs (1.73%), PUFAs (25.96%), CLAs (40.07%), n-3 (129.83%), and n-6 (14.56%) than the control milk, while the SO diet significantly reduced the amount of monounsaturated fatty acids (−13.49%) and the n-6/n-3 (−46.62%) ratio. In addition, the atherogenicity index determined in the diet enriched with FL was lower than the control diet due to a decrease in the saturated/unsaturated ratio.

Table 4 details the FA profiles of the fresh goat cheeses. There were significant differences (*p* < 0.05) in all the FAs determined, except C7:0 and C17:0, between the cheeses. The FA profile was similar to that observed in milk. Increases in the PUFA and CLA contents and decreases in the n-6/n-3 ratio and atherogenicity index compared with the control diet were observed, particularly in the cheeses made with milk from the animals receiving FL.

### 3.3. Texture and Sensory Profile

The results obtained for the texture profile (Table 5) indicated that no significant differences were observed in any of the parameters studied (*p* > 0.05), except adhesiveness (*p* < 0.05); the cheeses made with SO or FL milks were less adhesive than the control cheeses. Although no significant differences were found regarding hardness, the cheeses derived from the supplemented diets were less firm, probably because of the PUFA content.

In a milk sensory analysis conducted by a trained panel (Table 6), no significant differences were observed between the different milks for any of the determined sensory attributes (*p* > 0.05). However, although no significant differences were found, the milk with the highest overall acceptance was that obtained from the control group and the least acceptable was the milk from goats that were fed the SO diet.

Moreover, no significant differences were found in the sensory profile (*p* > 0.05) between the different types of cheeses (Table 7). However, the cheeses with the highest overall score were the controls; the lowest scores corresponded to those made with milk from animals given a diet supplemented with SO, reflecting the results obtained in milk (Table 6).

## 4. Discussion

### 4.1. Diet Intake and Physicochemical Milk/Cheese Composition

The slight decrease in the intake of the SO diet was not statistically significant. However, the studies carried out showed that feed consumption decreased by 6% in goats in response to a mixture of extruded linseeds and fish oil (representing 14.7% and 1.7% of the DM in the respective diets), while adding 21% extruded linseed alone did not affect the intake, although both diets had the same ether extract content (6.9% DM [24]. In goats, the inclusion of 3% unprotected fish oil reduces feed intake, but supplementation with protected fish oil has no effect on the same parameter [25]. Thus, the effect of adding fatty supplements on intake depends on many factors, including ruminant species, the amount included in the diet, and the type and composition of any supplement.

The dietary treatments had no significant effect on daily milk yield despite the substantial reduction in feed intake when a mixture of extruded linseeds and fish oil was included in the diet [24]. As shown in Table 2, the supplementation of goat diets with fats with a higher level of polyunsaturated fatty acid had no effect on the physicochemical parameters of the resulting milk. In goat milk, unlike in cow milk, there is no decrease in the milk protein and fat content when the goat diet is supplemented with PUFA-rich vegetable oils, which can partially be explained by the fact that the inhibition of acetyl-CoA carboxylase and de novo lipogenesis is less strong in the goat mammary gland [26]. However, significant differences were observed in the dry matter content of the milk, which was lower in the milk from animals receiving the SO diet than in the corresponding milk from FL diet, in agreement with studies carried out in goats [27] and in ewes fed diets supplemented with fish oil [28]; one possible explanation may be the hypophagic effect of long-chain PUFA from fish oil [9,29].

Regarding fresh goat cheese, no influence of either diet supplementation was observed on the physicochemical parameters when linseed supplementation was used [30,31]. Regarding the technological suitability for cheese-making, no significant differences were observed in the clotting time or cheese yield between diets. Milk clotting time is associated with the milk physicochemical composition, mainly its protein concentration; thus, as expected, no differences between milks were observed in this parameter [32]. The protein content, especially, would explain why no differences were found in the cheese yield.

Our results agree with those obtained in Padraccio cheese derived from a dietary supplementation with extruded linseed [33] where no significant differences were observed in any rheological characteristics. However, cheese texture derived from cows supplemented with extruded linseed and vitamin E, and plant extract produced a softer, more uniform, meltable, and fatty texture than in the control [34]; these results could be explained in reference to the lower fat melting point of the cheeses due to the higher PUFA content. In our study, although no significant differences were found regarding hardness, a softer texture was correlated to a higher unsaturation level [35]. However, an increase in hardness was observed with linseed supplements associated with the lower moisture of supplemented cheeses [31]. Changes in the textures of different types of fortified cheese could be explained by the interactions between milk components, enzymes, and sources of fat [36] and by the different technological treatments applied during cheese-making.

### 4.2. Fatty Acid Profile

Diets rich in n-3 FAs affect the FA composition not only by direct assimilation into milk, but also by modulating the expression of lipogenic enzymes [37]. Following the same pattern, as was described by several researchers of dairy cows, supplementation with sunflower oil decreased the SFA and increased the total n-3 FA. However, FL supplementation enhanced the n-3FAs, especially ALA, in the diet supplemented with FL, which would contribute to a decrease in cardiovascular disease risk factors due to reduced levels of serum low-density lipoprotein cholesterol. In addition, the FL diet decreased the levels of palmitic acid (C16:0) and hypercholesterolemic saturated acid [27]. The RA level was highest in the diet supplemented with FL, in accordance with the findings obtained in Manchega ewes when their diet was supplemented with extruded linseed [38]. The greater increase may have been due to the levels and form of the linseed because the extrusion process increases the accessibility of ALA to rumen microbiota. The resulting alteration of the rumen metabolism could make biohydrogenation less efficient and may also decrease the saturation ratio, increasing the concentration of C18:3 and trans-fatty acids in milk from diets rich in linseed oil [27]. The isomer trans-10 cis-12 (another CLA) always remains at trace levels in goat milk because this CLA is converted into C18:1, trans-10 in the rumen, and an increase in this FA was only observed when it was infused postruminally [39]. However, diets with extruded linseed had a minimal effect on this isomer, which agrees with our results [38]. On the other hand, a diet supplemented with fish oil increased the concentrations of CLA and long-chain PUFA and decreased C18:0 in milk, as also shown in our study [40].

Goat seems to be less sensitive than cows to the shift from C18:1, trans-11 to trans-10, which would explain the stability of the cis-9, trans-11 CLA determined in our study. However, increases in the percentages of C18:2, trans-9, and trans-12 were observed in the milk of animals receiving FL/SO diets, as supported in previous studies using diets supplemented with extruded linseed [38].

In our study, we found that the highest DHA values are associated with an increase in SFA concentration and a higher saturated/unsaturated fatty acid ratio, in contrast to that observed in milk derived from ewes supplemented with tuna oil [25]. In a study considering the influence of the type of diet in goat and ewe milk [27] suggested that the biohydrogenation of PUFAs in soybeans or linseed would occur slowly, producing SFA and less C18:1 or CLA; however, this effect was not observed in our study, possibly due to the way that the seeds had been treated, since the process used to obtain flaked linseed breaks the seed and increases the accessibility of ALA. Therefore, the diet supplemented with FL significantly improved the nutritional value of the subsequent milk due to the reduction in SFA and increased levels of PUFA and CLA isomers [4]. A diet rich in linseed oil decreased the n-6/n-3 ratio and significantly increased the CLA levels, which agrees with our results [41]. A nutritional improvement was also observed in milk from goats given the diet supplemented with SO due to the significant increases in PUFA and CLA. Although several studies have stated that the n-6/n-3 PUFA dietary ratio is of no relevance for modifying the risk of cardiovascular disease, there are studies which have determined that the conversion of long-chain omega-3 PUFAs (n-LCPUFAs), such as EPA and DHA, is reduced by a high ratio of linoleic/linolenic acids [42]. Thus, an increase in the dietary intake of pre-formed n-LCPUFA or reducing the n-6 PUFA intake is recommended, or a combination of both; however, direct DHA intake is more efficient. Indeed, this was the case with our results for the fatty acid profile and n-6/n-3 ratio of the milks from the diets enriched with flaked linseed or fish oil, and for the increase in DHA with the SO diet. In addition, the decreases in the atherogenicity index observed in milk and cheese resulting from the diet enriched with FL, due to decreases in the saturated/unsaturated ratio, confirmed the results obtained in goats using a diet supplemented with unsaturated plant oils and those obtained in grazing goats with a diet supplemented with extruded linseed [43]. A decrease was also observed in milk and cheese derived from Manchega ewes fed extruded linseed [38].

Unlike milk, few studies have investigated the effect of diet supplementation on FA profiles in cheese. The similar FA patterns observed in milk and cheese agree with the results determined by Gebreyowhans et al. [4]. The FA profiles of cheeses, mainly FAs associated with potential benefits to human health, primarily depend on the FA composition of the milk used rather than the cheese-making technology [44]. 

It should be highlighted that none of the ALA values obtained in our study for milk or cheese derived from an FL-supplemented diet were higher than the overall mean value found across European countries [45]. 

The CLA determined in cheese was observed to be primarily dependent on the CLA level of the unprocessed milk. In our study, increases compared with the control were observed in the PUFA and CLA contents, as were decreases in the n-6/n-3 ratio and atherogenicity index, particularly in the cheeses made with milk from the animals receiving FL. Notably, although our supplementation-control periods were short, significant changes in the fatty acid profile were evident, which indicated that it is possible to change the nutritional value of animal products using short time intervals, such as that defined in our trial. However, longer studies should be conducted to evaluate their effects on animal health or further changes in the nutritional value of products.

### 4.3. Milk/Cheese Sensory Profile

No significant differences were observed by the trained panel between the milk and fresh goat cheese resulting from the different diets in any of the determined sensory attributes. These results confirm that it is possible to obtain milk/cheese with better nutritional characteristics without altering the sensory profile. These results agree with those obtained by Dauber et al. [46] in goat cheese derived from milk supplemented with sunflower oil. Our results are in agreement with those observed by Nguyen et al. [44] in ewe cheese, although they observed that levels of MUFA exerted a strongly negative effect on cheese edible quality, which can partially explain the lowest overall acceptance of cheeses derived from the SO diet. However, in commercial CLA-fortified dairy products, some defects or losses in flavor have been determined [47]. Differences were determined in Pecorino cheese odor, flavor, and toughness as a result of a diet supplemented with extruded linseed; lower odor and higher toughness and flavor values were found for a CLA-enriched cheese [48]. Thus, in our study, although no significant differences were found, the highest overall acceptance was obtained from the milk and cheese corresponding to the control group in accordance with the results observed by Santurino et al. [30], where the least acceptable was the milk/cheese from goats given the SO diet.

Furthermore, this study was designed as practical applied research in nutrition; thus, the omega-3-rich supplements selected for evaluation were commercially available flaked linseed rich in ALA (Agrocava S.L., Caravaca de la Cruz, Murcia, Spain), and a salmon fish oil product rich in DHA and EPA (Optomega-50, Optivite International Ltd., Nottinghamshire, UK). Therefore, the milk obtained from animals receiving dietary supplementation can be provided to the dairy industry, mainly goat production.

## 5. Conclusions

This study shows that a diet supplemented with FL or SO modifies the FAs of milk and cheese of dairy goats, with marginal effects on the physicochemical composition. Therefore, for both milk and cheese, dietary supplementation, especially with FL, results in a product with a higher nutritional quality than that obtained using the diet routinely administered on farms, especially as far as the fatty acid profile is concerned. Based on the sensory analysis conducted in milk and fresh cheese from dairy goats in this study, no significant differences existed between the control and supplemented groups, which is important from the consumer’s point of view because any increase in price because of a product being healthier must be justified by sensory properties, which should equal to or better than those provided by the traditional product.

## Figures and Tables

**Table 1 animals-13-03652-t001:** Ingredients and chemical composition of experimental diets.

	Control	Flaked Linseed	Fish Oil
Ingredients (% of DM)			
Alfalfa hay	16.92	16.92	16.92
Citrus pulp	15.86	15.77	15.77
Soaps of palm oil fatty acids ^1^	1.55		
Flaked linseed ^2^		3.88	
Fish oil concentrate ^3^			2.64
Concentrate mixture ^4^	65.67	63.43	64.67
Nutritive value calculated ^5^			
UFL ^6^/kg DM	1.01	1.01	1.01
CP, % of DM	16.58	17.01	16.52
PDIE ^7^, % of DM	10.8	10.3	10.3
PDIN ^8^, % of DM	10.3	11.0	10.7
PDIA ^9^, % of DM	4.6	4.7	4.6
Ether extract, % of DM	4.5	4.5	4.5
Chemical composition analyzed			
DM, %	57.84	57.97	57.79
CP, % of DM	15.50	16.38	16.17
Ether extract, % of DM	4.01	5.21	4.18
NDF, % of DM	37.17	37.70	37.67
ADF, % of DM	21.52	22.46	22.56
ADL, % of DM	3.39	3.41	3.74
Starch, % of DM	20.01	20.00	20.00
Fatty acids (g/100 g of total FA)			
C14:0	1.91	1.38	3.31
C16:0	20.20	16.49	18.35
C16:1	1.69	1.20	3.47
C18:0	9.11	8.16	9.60
C18:1	31.41	17.2	20.80
C18:2n-6	27.00	21.56	26.57
C18:3n-3	6.97	32.71	14.43
C20:4n-6	0.00	0.00	0.13
C20:5n-3	0.00	0.00	0.09
C22:6n-3	0.00	0.00	0.10
Saturated	31.23	26.04	31.27
Monounsaturated	33.10	18.42	24.27
Polyunsaturated	33.97	54.27	41.32

^1^ Calcium soaps of palm oil fatty acids (84% EE) (MAGNAPAC^®^ NOREL SA, Madrid, Spain). ^2^ Linseed flakes (34% EE). ^3^ Fish oil concentrate (salmon source) (OPTOMEGA-50, Optivite, International Ltd., Barcelona, Spain) (50% EE). ^4^ Concentrate mixture containing barley grain (233.1 g/kg), sunflower meal (165.70 g/kg), soybean hulls (225.50 g/kg), wheat grain (40.11 g/kg), dehydrated sugar beet pulp (25.49 g/kg), corn grain (103.25 g/kg), soybean meal (37.38 g/kg), corn gluten feed (121.21 g/kg), urea (4.87 g/kg), animal fat (lard) (7.83 g/kg), dicalcium phosphate (6.20 g/kg), calcium carbonate (8.88 g/kg), sodium bicarbonate (4.93 g/kg), magnesium oxide (2.87 g/kg), and 12.68 g/kg of trace minerals and vitamin supplements, containing copper (900 mg/kg), manganese (6000 mg/kg), zinc (5000 mg/kg), vitamin A (800,000 IU/kg), vitamin D (175,000 IU/kg), and vitamin E (6000 IU/kg). ^5^ According to FEDNA (2010). ^6^ UFL: forage unit for milk production (1 UFL = 7.1128 MJ of net energy for lactation). ^7^ PDIE: protein digestible in the intestine, with energy as the limiting factor for rumen microbial growth. ^8^ PDIN: protein digestible in the intestine, with nitrogen as the limiting factor for rumen microbial growth. ^9^ PDIA: protein digestible in the intestine supplied by rumen-undegraded dietary protein.

**Table 2 animals-13-03652-t002:** Influence of diet on dry matter intake, yield milk and the physicochemical parameters of goat milk.

	Control	Flaked Linseed	Fish Oil	SEM ^1^	*p*-Value ^2^
DM intake (kg/animal and day)	2.07 ^a^	2.02 ^ab^	1.99 ^b^	0.010	†
Milk yield (kg/animal and day)	2.42	2.46	2.47	0.110	ns
pH	6.69	6.68	6.68	0.037	ns
Dry matter (g/100 g)	13.54 ^ab^	13.75 ^a^	13.34 ^b^	0.373	*
Protein (g/100 g)	3.14	3.14	3.10	0.039	ns
Fat (g/100 g)	5.28	5.35	5.03	0.241	ns

^1^ SEM = standard error of the mean (n = 3). ^2^ Probability of significance: ns, *p* > 0.05; †, *p* < 0.1; *, *p* < 0.05. ^ab^ Means with different letters indicate differences (*p* < 0.05) between columns.

**Table 3 animals-13-03652-t003:** Influence of diet on the physicochemical parameters of fresh goat cheese.

	Control	Flaked Linseed	Fish Oil	SEM ^1^	*p*-Value ^2^
pH	6.88	6.90	6.89	0.019	ns
Dry matter (g/100 g)	41.27	41.22	41.18	1.304	ns
Protein (g/100 g)	13.77	13.94	14.11	0.320	ns
Fat (g/100 g)	24.71	25.11	22.87	1.143	ns
Coagulation time (T_max_)	7.42	7.70	7.50	0.217	ns
Yield (L/Kg)	5.40	5.33	5.47	0.098	ns

^1^ SEM = standard error of the mean (n = 3). ^2^ Probability of significance: ns (non-significant), *p* > 0.05.

**Table 4 animals-13-03652-t004:** Influence of diet on the fatty acid profiles of goat milk and fresh goat cheese (mg/100 g serving of milk or cheese).

	Milk					Cheese				
	Control	Flaked Linseed	Fish Oil	SEM ^1^	*p*-Value ^2^	Control	Flaked Linseed	Fish Oil	SEM ^1^	*p*-Value ^2^
C4:0	67.06	69.02	62.88	0.018	ns	306.40 ^a^	311.36 ^a^	263.01 ^b^	0.047	***
C6:0	112.99 ^b^	121.45 ^a^	115.69 ^a^	0.056	***	111.41 b	117.17 ^a^	106.64 ^b^	0.078	*
C7:0	1.06 ^c^	1.61 ^b^	2.01 ^a^	0.001	***	1.58	1.61	1.51	0.001	ns
C8:0	157.87 ^c^	173.34 ^b^	172.03 ^a^	0.070	***	154.18 ^b^	171.74 ^a^	162.97 ^a^	0.080	***
C9:0	3.70 ^c^	4.28 ^b^	4.53 ^a^	0.001	***	3.70 ^c^	4.28 b	4.53 ^a^	0.002	***
C10:0	571.30 ^c^	615.25 ^b^	630.26 ^a^	0.120	***	562.85 ^c^	612.04 ^b^	603.60 ^a^	0.085	***
C12:0	252.38 ^c^	265.36 ^b^	296.77 ^a^	0.090	***	250.27 ^c^	269.11 ^b^	286.71 ^a^	0.080	***
C13:0	6.34 ^b^	6.96 ^b^	7.04 ^a^	0.004	***	6.34 ^c^	6.96 ^b^	7.04 ^a^	0.005	***
C14:0	459.36 ^c^	481.50 ^b^	509.54 ^a^	0.138	***	458.30 ^c^	480.97 ^b^	505.01 ^a^	0.102	***
C14:1	7.92 ^b^	7.49 ^c^	9.56 ^a^	0.005	***	7.39 ^b^	7.49 ^b^	9.56 ^a^	0.011	***
C15:0	42.24 ^c^	45.48 ^b^	48.79 ^a^	0.016	***	43.30 ^c^	45.48 ^b^	48.29 ^a^	0.007	***
C16:0	1870.70 ^a^	1679.90 ^c^	1732.84 ^b^	0.371	***	1895.52 ^a^	1691.14 ^c^	1749.94 ^b^	0.481	***
C16:1	46.99 ^b^	39.59 ^c^	59.86 ^a^	0.020	***	48.05 ^b^	41.20 ^c^	60.86 ^a^	0.048	***
C17:0	27.98 ^c^	29.43 ^b^	29.68 ^a^	0.011	***	27.98	28.89	28.67	0.019	ns
C18:0	417.65 ^b^	495.95 ^a^	263.07 ^c^	0.309	***	417.12 ^b^	481.50 ^a^	285.70 ^c^	0.547	***
C18:1	930.86 ^a^	929.83 ^a^	714.26 ^b^	0.225	***	922.94 ^a^	916.99 ^b^	729.85 ^c^	0.300	***
C18:1, cis-11	25.34 ^b^	25.15 ^b^	37.73 ^a^	0.027	***	26.40 ^b^	25.15 ^b^	37.73 ^a^	0.068	***
C18:2, n6	168.43 ^c^	203.30 ^a^	184.10 ^b^	0.157	***	165.79 ^c^	200.09 ^a^	180.07 ^b^	0.035	***
C18:2, trans-9. trans-12	12.67 ^c^	30.50 ^a^	15.59 ^b^	0.016	***	13.20 ^c^	29.43 ^a^	15.09 ^b^	0.018	***
C18:2, cis-9. trans-11	34.85 ^c^	51.90 ^a^	45.27 ^b^	0.036	***	38.02 ^c^	51.90 ^a^	47.79 ^b^	0.097	***
C18:2, trans-10, cis-12	0.53 ^c^	1.07 ^b^	1.51 ^a^	0.003	***	1.58 ^b^	2.14 ^a^	1.51 ^b^	0.010	***
C18:3	14.78 ^c^	52.97 ^a^	23.64 ^b^	0.014	***	14.78 ^c^	51.90 ^a^	24.14 ^b^	0.036	***
C19:0	1.58 ^b^	3.21 ^a^	2.01 ^b^	0.003	***	2.11 ^c^	3.21 ^a^	2.52 ^b^	0.003	***
C20:0	10.03 ^b^	10.17 ^b^	10.06 ^a^	0.003	**	10.03 ^b^	9.63 ^c^	10.06 ^a^	0.007	***
C20:1	5.81 ^b^	5.35 ^c^	15.59 ^a^	0.011	***	5.81 ^b^	5.35 ^b^	16.10 ^a^	0.023	***
C20:2 n6	3.17 ^c^	3.75 ^b^	5.03 ^a^	0.003	***	3.70 ^c^	3.75 ^b^	5.03 ^a^	0.005	***
C20:4 n6	19.01 ^b^	17.66 ^c^	21.63 ^a^	0.027	***	19.01 ^b^	17.12 ^c^	20.12 ^a^	0.022	***
C20:5 n3	1.58 ^c^	2.68 ^b^	4.53 ^a^	0.014	***	1.58 ^c^	2.68 ^b^	4.53 ^a^	0.015	***
C22:0	2.64 ^b^	2.68 ^b^	3.52 ^a^	0.003	***	2.11 ^b^	2.68 ^b^	4.02 ^a^	0.002	***
C22:5 n3	4.75 ^c^	6.42 ^b^	11.07 ^a^	0.017	***	4.22 ^c^	6.42 ^b^	10.56 ^a^	0.020	***
C22:6 n3	0.00 ^b^	0.00 ^b^	8.05 ^a^	0.059	***	0.00 ^b^	0.54 ^b^	10.56 ^a^	0.047	***
Saturated	4004.35 ^b^	3981.47 ^c^	3881.15 ^a^	0.189	**	4010.16 ^b^	3990.57 ^c^	3861.53 ^a^	0.283	***
Monounsaturated	1016.40 ^a^	1006.87 ^b^	836.49 ^c^	0.187	***	1011.12 ^a^	996.17 ^b^	854.09 ^c^	0.286	***
Polyunsaturated	223.87 ^c^	308.70 ^a^	269.11 ^b^	0.190	***	220.70 ^c^	311.37 ^a^	265.58 ^b^	0.103	***
CLA	34.85 ^c^	52.97 ^a^	46.78 ^b^	0.040	***	38.02 ^b^	52.43 ^a^	48.79 ^a^	0.095	***
n3	20.59 ^c^	62.60 ^a^	45.27 ^b^	0.070	***	20.06 ^c^	60.99 ^a^	47.28 ^b^	0.087	***
n6	190.61 ^b^	216.14 ^a^	208.24 ^a^	0.200	**	187.97 ^c^	220.42 ^a^	202.71 ^b^	0.088	***
n6/n3	9.26 ^a^	3.455 ^c^	4.59 ^b^	0.550	***	9.37 ^a^	3.61 ^c^	4.29 ^b^	0.467	***
Atherogenicity index	10.03 ^b^	157.83 ^c^	186.11 ^a^	0.080	***	170.54 ^b^	158.90 ^c^	182.59 ^a^	0.044	***
Saturated:unsaturated (ratio)	12.14 ^b^	162.11 ^c^	177.06 ^a^	0.040	***	172.13 ^b^	163.71 ^c^	173.54 ^a^	0.045	***

^1^ SEM: standard error of the mean (n = 3). ^2^ Probability of significance: ns (not-significant) *p* > 0.05; * *p* ≤ 0.05; ** *p* ≤ 0.01; *** *p* ≤ 0.001. ^a,b,c^ Means with different letters indicate differences (*p* ≤ 0.05) between columns for products.

**Table 5 animals-13-03652-t005:** Influence of diet on the texture profile analysis (TPA).

	Control	Flaked Linseed	Fish Oil	SEM ^1^	*p*-Value ^2^
Hardness 1	8.51	7.76	7.96	0.185	ns
Hardness 2	4.67	4.23	4.77	0.325	ns
Cohesiveness	0.36	0.37	0.39	0.029	ns
Gumminess	3.02	2.85	3.18	0.239	ns
Chewiness	35.26	33.48	37.26	2.871	ns
Adhesiveness	0.97 ^a^	0.74 ^b^	0.66 ^b^	0.086	*
Springiness	11.65	11.72	11.72	0.153	ns

^1^ SEM = standard error of the mean (n = 3). ^2^ Probability of significance: ns (not significant), *p* > 0.05; *, *p* < 0.05. ^ab^ Means with different letters indicate differences (*p* < 0.05) between columns.

**Table 6 animals-13-03652-t006:** Influence of diet on the sensorial parameters of milk.

	Control	Flaked Linseed	Fish Oil	SEM ^1^	*p*-Value ^2^
Whiteness	9.87	9.77	9.77	0.079	ns
Goaty odour	8.99	9.05	9.07	0.318	ns
Foreign odours	0.02	0.08	0.06	0.030	ns
Goaty taste	8.94	8.77	8.87	0.377	ns
Foreign flavours	1.09	1.05	1.20	0.278	ns
Fatty sensation	3.12	3.07	3.20	0.285	ns
Overall score	7.82	7.60	7.38	0.483	ns

^1^ SEM = standard error of the mean (n = 3). ^2^ Probability of significance: ns (not significant), *p* > 0.05.

**Table 7 animals-13-03652-t007:** Influence of diet on the sensorial parameters of cheese.

	Control	Flaked Linseed	Fish Oil	SEM ^1^	*p*-Value ^2^
Whiteness	9.61	9.61	9.47	0.202	ns
Goaty odour	0.90	1.03	0.98	0.283	ns
Foreign odours	0.03	0.08	0.06	0.038	ns
Goaty taste	1.37	1.77	1.65	0.342	ns
Foreign flavours	0.08	0.15	0.18	0.055	ns
Hardness	2.59	2.52	2.57	0.328	ns
Fatty sensation	1.76	1.64	1.63	0.263	ns
Overall score	7.30	7.28	7.15	0.374	ns

^1^ SEM = standard error of the mean (n = 3). ^2^ Probability of significance: ns (not significant), *p* > 0.05.

## Data Availability

None of the data have been deposited in an official repository; the data are available from the corresponding author upon reasonable request.

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
