# Peer review of "Influence of Dietary Lipid Source Supplementation on Milk and Fresh Cheese from Murciano-Granadina Goats"

_animals, 2023, doi:10.3390/ani13233652_

Round 1

Reviewer 1 Report

Comments and Suggestions for Authors

The article is of interest to producers of goat milk and products from it with improved quality characteristics for human nutrition, in particular, with a lower content of saturated fatty acids. The authors used adequate research methods on a large group of goats. Modern methods of chemical analysis have been applied. For some improvement of the content of the article there is some wish for additions.

It is desirable to provide a justification for why the authors chose the level of introduction into the diet of goats flaked linseed (FL) and salmon (fish) oil (SO) 3.88 or 2.64 % of diet on a dry basis, respectively. Explain why calcium soaps of palm oil is a control. To judge the economic feasibility of obtaining goat milk when using rations with the inclusion of calcium soaps of palm oil, flaked linseed and fish oil, add data on their cost and availability for farmers.

Author Response

The article is of interest to producers of goat milk and products from it with improved quality characteristics for human nutrition, in particular, with a lower content of saturated fatty acids. The authors used adequate research methods on a large group of goats. Modern methods of chemical analysis have been applied. For some improvement of the content of the article there is some wish for additions.

It is desirable to provide a justification for why the authors chose the level of introduction into the diet of goats flaked linseed (FL) and salmon (fish) oil (SO) 3.88 or 2.64 % of diet on a dry basis, respectively. Explain why calcium soaps of palm oil is a control. To judge the economic feasibility of obtaining goat milk when using rations with the inclusion of calcium soaps of palm oil, flaked linseed and fish oil, add data on their cost and availability for farmers.

The choice of the incorporation level of salmon fish oil and flaked linseed was based on obtaining rations with a similar energy (and EE, level formulated 4.45%, Table 1) and protein concentration, capable of covering the requirements of the goats, without exceed the EE content of diets. Furthermore, our trial focused on diets that could be acceptable under commercial conditions. On the other hand, at a commercial level, it is very common to use palm calcium soaps in high production goats, an ingredient with high content of saturated fatty acids. Thus, it was the control chosen to compare its effect on animal products against those obtained with other ingredients rich in omega-3 fatty acids (such as flaked linseed and salmon fish oil). In addition, the percentages of incorporation of the supplements are different, since the lipid content differs between them (calcium soaps of palm oil FA, an ingredient with a 84% of ether extract; flaked linseeds, seeds with a 34.7% of ether extract; and Salmon oil, a product with 50% of ether extract). An aspect considered for the diet formulation. Now we have incorporated the ether extract content of these ingredients into the description of the materials and methods.

Regarding the costs and availability of these ingredients, these are dependent on the raw materials market, and the ability to generate an attractive product for the consumer, so the farmers will be able to resort to them if they find an economic benefit.

Reviewer 2 Report

Comments and Suggestions for Authors

Dear authors. thanks for your manuscript. The number of goats is remarkable

A modification of english is, in my opinion, mandatory. The grammar quality of your manuscript is below average and difficult to understand, especially the first sections.

Other general comments 

Although a remarkable number of goats was used in the present work, the duration of the trial was short and probably debatable to relate diet modifications in the production and nutritional quality of goat’ milk and cheese. My recommendation would be to add in the discussion the implications, strengths, and weakness that possess short-term trials like this.

The abstract does not have neither introduction nor conclusion. It is mainly dedicated to methodology and results. This might be modified

The title is not self-explanatory of this manuscript

The introduction has only three references. This is a low number given the vast of knowledge produced for goat nutrition and products quality.

Information in lines 63 to 65 is more likely to be established in M&M

Information in lines 65 to 68 is more likely to be established in the abstract or discussion.

Tables are not in a proper format. They see like copied and pasted for another Microsoft word or adobe reader document. Aesthetically and according to the editorial guidelines, is unproper.

In line 43 it is difficult to understand which strategy authors are referring

Authors mentioned “these studies” albeit only one was cited: Gebreyowhans et al., 2019

I must disagree with the phrase between lines 59-60: “Nowadays it is technically feasible to modify FAs by feeding strategies, there is a gap of knowledge about the effects of such supplementation on the properties of the resulting milk and cheese need to be further investigated and constitute the proposal of this study”.  Worldwide there is plenty if information on this topic both for sheep and goats.

According with the above exposed, the amount of literature related with supplementation strategies to improve the quality of goat dairy products is quite extensive and performed in many agroecological conditions and production systems. Thus, the introduction of this article provides the idea of a lack of knowledge regarding this topic. I consider that this is not the proper strategy to elaborate the introduction.

Please consider to better explain the experimental design a more easy interpretation by readers.

It would be fine if authors add information on the experimental animals. This is weight, number of lactations, age, or another that can help verify the homogeneity amongst experimental groups.

More information on the location and general characteristics of the productive system is needed

There is no information or references for the organoleptic panel. It seems somewhat arbitrary. Please specify  – and reference – how this methodology  was performed.

L90-91: The diets cover the nutritional requirements for goats, but according to what author, organization, feeding systems??. Please reference and add the content of nutrients that goats were eating

L97-98: The processes/methodologies employed to assess the content of dry matter and chemical analyses of feedstuffs are important and need to be specified in the M&M section.

L105-107 The process to obtain cheese (according with the methodology of Garcia et al., 2012) has to be briefly explained

Components of the formulae in L129-131 need to be specified and referenced.

The intake per day, on an individual animal basis, should be avoided/excluded since the measurements were performed in a group manner.  

Comments on the Quality of English Language

it needs a modification of the grammar quality

Author Response

A modification of english is, in my opinion, mandatory. The grammar quality of your manuscript is below average and difficult to understand, especially the first sections.

The article was revised by MDPI English editing service.

Other general comments 

Although a remarkable number of goats was used in the present work, the duration of the trial was short and probably debatable to relate diet modifications in the production and nutritional quality of goat’ milk and cheese. My recommendation would be to add in the discussion the implications, strengths, and weakness that possess short-term trials like this.

This paragraph has been included in the discussion.

“It should be noted that although our supplementation-control periods were short, significant changes in the fatty acid profile were evident, which indicated that it would be possible to change the nutritional value of animal products using the short time interval defined in our trial. However, longer studies should be conducted to evaluate their effects on animal health or further changes in the nutritional value of products.

The abstract does not have neither introduction nor conclusion. It is mainly dedicated to methodology and results. This might be modified.

Revised and corrected.

The title is not self-explanatory of this manuscript.

The title was changed to a more self-explanatory one.  (Influence of dietary lipid source supplementation on milk and fresh cheese of Murciano-Granadina goats)

The introduction has only three references. This is a low number given the vast of knowledge produced for goat nutrition and products quality.

Thanks for their comments, the authors have enriched the introduction with further bibliographical references.

Information in lines 63 to 65 is more likely to be established in M&M.

These lines are deleted.

Information in lines 65 to 68 is more likely to be established in the abstract or discussion.

These sentences are included in the abstract section.

Tables are not in a proper format. They see like copied and pasted for another Microsoft word or adobe reader document. Aesthetically and according to the editorial guidelines, is unproper.

The tables are provided in a different file; however, tables are modified to improve its quality.

In line 43 it is difficult to understand which strategy authors are referring.

This paragraph is rewritten for better understanding.

Authors mentioned “these studies” albeit only one was cited: Gebreyowhans et al., 2019.

 Modified

I must disagree with the phrase between lines 59-60: “Nowadays it is technically feasible to modify FAs by feeding strategies, there is a gap of knowledge about the effects of such supplementation on the properties of the resulting milk and cheese need to be further investigated and constitute the proposal of this study”.  Worldwide there is plenty if information on this topic both for sheep and goats.

Specified in the text that it is due to Murciano-Granadina goats

According with the above exposed, the amount of literature related with supplementation strategies to improve the quality of goat dairy products is quite extensive and performed in many agroecological conditions and production systems. Thus, the introduction of this article provides the idea of a lack of knowledge regarding this topic. I consider that this is not the proper strategy to elaborate the introduction.

Modified

Please consider to better explain the experimental design a more easy interpretation by readers.

The experimental design is better explained.

It would be fine if authors add information on the experimental animals. This is weight, number of lactations, age, or another that can help verify the homogeneity amongst experimental groups.

The average body weight has been added ± standard deviation: “(average body weight of 44.65 ± 3.48 kg)”

More information on the location and general characteristics of the productive system is needed We have this information but in order of better understanding we decided not included in the present manuscript.

Production system and location have been indicated: “The experiment was conducted in a farm under an intensive production system, located in the southeast of Spain (Kpra Leche y Genética, S.L., Mula, Murcia, Spain).”

There is no information or references for the organoleptic panel. It seems somewhat arbitrary. Please specify  – and reference – how this methodology  was performed.

More information about the sensory analysis is provided.

L90-91: The diets cover the nutritional requirements for goats, but according to what author, organization, feeding systems??. Please reference and add the content of nutrients that goats were eating

This paragraph has been completed and improved:

All diets cover the nutritional requirements for lactating goats according the Fundación Española para el Desarrollo de la Nutrición Animal (FEDNA) [2010], and they were formulated to be iso-energetic and iso-nitrogenous.”

In addition, in Table 1 the information of energy and protein content of the dietary diet is shown.

L97-98: The processes/methodologies employed to assess the content of dry matter and chemical analyses of feedstuffs are important and need to be specified in the M&M section.

Included a new section about feed analysis.

L105-107 The process to obtain cheese (according with the methodology of Garcia et al., 2012) has to be briefly explained.

The paragraph is rewritten.

Components of the formulae in L129-131 need to be specified and referenced.

The formula is referenced.

The intake per day, on an individual animal basis, should be avoided/excluded since the measurements were performed in a group manner.  

We think that since the intake by group and dietary treatment, as well as the animals per group were recorded in each control period. Moreover, the dietary treatments were offered alternatively to all groups, thus we could calculate this data, what we consider very valuable and difficult to obtain under a farm condition trial.

Reviewer 3 Report

Comments and Suggestions for Authors

notes in the attached file

Author Response

1.the title says about a diet -please specify which one.

 The title is modified.

2.please remove double spaces for example in line 37.

Deleted.

3.please correct the designation of DHA on line 42 and throughout the paper.

Modified.

  1. in the paragraph on lines 37-42 the authors refer to numerous studies, please specify which ones.

Modified.

  1. please standardize or use salmon oil (SO) or fish oil.

Standarized.

  1. line 43 -what strategy do the authors have in mind.

Rewritten.

  1. it is known that the responses regarding goat milk production and mammary lipid metabolism are different from those observed in cow-Please confirm the credibility of this statement with scientific research

This paragraph has been improved and completed:

However, most of these studies were performed in dairy cow, but it has suggested that there are differences in lipid biohydrogenation pathways in the rumen, and in lipogenesis in the mammary gland between cows and goats (Toral et al., 2015). Thus, Bernard et al. (2017) found that among 19 mRNAs encoded by genes related to lipid metabolism in the mammary gland, 15 of them had a different abundance (9 were more abundant in cows and 6 in goats). Furthermore, Fougère et al. (2018) showed that the effect of different lipid sources could have different consequences on the lipid profile or fat content of milk, and the profile was also dependent of the ruminant species (goats or cows).

Reference: Toral, P. G., Chilliard, Y., Rouel, J., Leskinen, H., Shingfield, K. J., & Bernard, L. (2015). Comparison of the nutritional regulation of milk fat secretion and composition in cows and goats. Journal of dairy science, 98(10), 7277-7297.

Reference: Bernard, L., Toral, P. G., & Chilliard, Y. (2017). Comparison of mammary lipid metabolism in dairy cows and goats fed diets supplemented with starch, plant oil, or fish oil. Journal of dairy science, 100(11), 9338-9351.

Reference: Fougère, H., Delavaud, C., & Bernard, L. (2018). Diets supplemented with starch and corn oil, marine algae, or hydrogenated palm oil differentially modulate milk fat secretion and composition in cows and goats: A comparative study. Journal of dairy science, 101(9), 8429-8445.

  1. please provide the approval number of the ethics committee.

I enclose the certificate of the ethic committee signed that no assessment from this committee is necessary.

  1. please include in the list FEDNA 2010 publications.

 Included.

  1. Please perform an analysis of the correlation between the content of fatty acids in feed and in milk. The authors are investigating the effect of different oils that change the fatty acid profile of the feed on the milk profile. It is obvious that the more of a given fatty acid will be in the feed, the more it will be in the milk -where is the innovation here?

We think that in monogastric animals a very important correlation could be expected between the fatty acid profile of the diet and the composition of their animal tissues or products. However, in ruminants this effect is not direct, due to the action of ruminal microorganisms; since these can alter the lipid profile before fatty acids are absorbed. Furthermore, these effects can also be mediated by multiple factors such as the type of fatty acids, the form of presentation, the lipid level of the diet, etc. In this sense, the comparison of the lipid profile of milk produced by goats fed with different lipid sources (in line with our design), provides scientific information of relevance in applied feeding. So, in consequence, a correlation could not provide more information.

As a novelty, we have indicated and highlighted in the discussion that: “It should be noted that although our supplementation-control periods were short, significant changes in the fatty acid profile were evident, which indicated that it would be possible to change the nutritional value of animal products using the short time interval defined in our trial.”

  1. in the publication, it is not necessary to specify the differences if they are not significant -this is obvious (example line 149).

Modified.

12.i n paragraphs 176-187, the authors write about CLA and PUFA, it is worth indicating which acids the authors have in mind and PUFA corresponded to those determined by the authors in the present study included in

The results of the CLA of this article included, cis-9, trans-11 C18:2 and , trans-10, cis-12, C18:2 and the following PUFA acids are considered (C18:2 n-6, trans-9, trans-12 C18:2).

  1. the data style in the table is different than in the entire publication - correct it.

The style and format are corrected.

  1. please describe in detail the methodology of cheese production.

Included

  1. please merge tables 4 and 5 into one.

Tables 4 and 5 have been unified. Please see table 4

  1. Please describe in detail the methodology for texture and sensory analysis.

A new paragraph is included.

  1. please provide a summary in your work.

Included.

Reviewer 4 Report

Comments and Suggestions for Authors

The manuscript (animals-02570136) investigates the effects of different dietary fat source supplementation on milk and fresh cheese of Murciano-Granadina goats. The content of this manuscript totally falls within the scope of the special issue: Alternative Forages, Novel Feeds, and Supplementation Effects on Animal Performance and Product Quality: Meat, Milk, and Natural Fibres of Animals. To the best of my knowledge this paper has not been published elsewhere. The authors have investigated an interesting research field. They have done a huge and complicated experiment with 350 milking goats in a commercial scale. The manuscript was relatively well-prepared. After carefully reviewing the manuscript, I feel that it will be suitable for publication in this journal with a major revision. I have several comments for the authors to consider as follows:

General comments:

The author should carefully read and follow the instruction for authors.

Please use the full form of the words at the first time instead of its abbreviation and use full word forms at the beginning of sentences.

Do not need to use the full-form words after abbreviating.

Please be consistent in using abbreviations throughout the paper. I can see n-3, n-3FAs, n-3 FAs, n-3 FA; or C18:3 and C18:3 n-3.

Please be consistent in using only American English or British English throughout the paper.

There are a number of typos and spelling mistake.

Several sentences are unclear. They need to rewrite in a clear and concise way, or split them into different ones.

Some information in the manuscript need to provide at least a reference.

Some statements regarding the present study’s results are subjective and statistically incorrect.

Several cited references are not relevant. Please check again.

Specific comments:

Title:

I suggest replacing “Diets Supplemented” by “Dietary Lipid Source Supplementation”.

Affiliation:

Provide affiliations of I. Peñaranda, M.D. Garrido; and number the affiliations in the author list in an increasing order from the left to the right.

A simple summary:

 Please provide a simple summary before the abstract section.

Abstract:

Lines 14-15: Please provide % of supplements.

Line 23: add “fatty” between “polyunsaturated” and “acids”.

The authors should a brief take-home message of the study at the end of the abstract.

Introduction:

Lines 39-42: Need at least a reference

Please provide some more information about linseed and salmon oil. Why did you choose them to become the study’s supplements?

Line 61: Need at least 2 references

Materials and Methods:

Line 89: Why did the authors choose those different supplementing levels?

Lines 90-91: Which nutritional standards or system did you follow to state that all diets are the nutritional requirement? Please provide reference. What were the energy and protein density of the diets?

Lines 91-92: In table 1, the energy values of the diets were not found. Additionally, %CP were not similar among the treatments. As a result, the experimental diets seemed to be not isocaloric and isonitrogenous. Furthermore, the %EE are different among the diets. This may lead to unreliable results of the study. The authors may not be able to explain which factors actually affects their results. Please revise them and provide reasonable explanations.

Line 100-135: Although there are cited references, the sample collection and measurement methods need to have brief description. In the current version, they are too abstract.

Some equipment (e.g. a CoAguLiteTM optical sensor…) need to have the name of produced company, city and country. The information needs to be consistent in all of the equipment.

Results:

Lines 206-208 and 211-214: Those statement are statistically incorrect. Please remove them.

Please make a,b,c in the tables superscript.

Line 159: the unit of the clotting time in Table 3 is hour not Tmax

Discussion:

Lines 228-229: This is not correct. There was a tendency to reduce (p=0.1, Table 2), but not substantially reduced (p<0.05).

Lines 234-235: I could not find this information in the publication of Ferlay and Chillard (2020). Please check again.

Line 254: Please remove: “less firm and”.

Lines 257-258: This is a subjective statement. Please avoid.

Line 264: Please provide a reference.

Lines 337-340: Please split this sentence into 2.

Lines 342-343 and 348-351: They are not statistically correct.

Conclusion:

This section has not been found in the current version. I suggest the author separate the last paragraph of the Discussion section and make it become the conclusion setrion.

Financial Support Statement:

Please provide the address of the financial sponsors.

Ethics approval:

I am concerned about the animal welfare in this experiment. Although the farm staff directly take care of the animals as normal, animal selection, the treatment assignments, changing in diets… might have effects on the animal welfare, which the farm staff could not professionally recognise and solve.

Authors Contributions:

Please follow the journal instruction, the author initials only.

Reference list:

Please follow the author guideline of the journal.

Comments on the Quality of English Language

Please be consistent in using only American English or British English throughout the paper. There are a number of typos and spelling mistake.

Author Response

The manuscript (animals-02570136) investigates the effects of different dietary fat source supplementation on milk and fresh cheese of Murciano-Granadina goats. The content of this manuscript totally falls within the scope of the special issue: Alternative Forages, Novel Feeds, and Supplementation Effects on Animal Performance and Product Quality: Meat, Milk, and Natural Fibres of Animals. To the best of my knowledge this paper has not been published elsewhere. The authors have investigated an interesting research field. They have done a huge and complicated experiment with 350 milking goats in a commercial scale. The manuscript was relatively well-prepared. After carefully reviewing the manuscript, I feel that it will be suitable for publication in this journal with a major revision. I have several comments for the authors to consider as follows:

General comments:

The author should carefully read and follow the instruction for authors.

The article was revised by MDPI English editing service.

Please use the full form of the words at the first time instead of its abbreviation and use full word forms at the beginning of sentences.

Corrected

Do not need to use the full-form words after abbreviating.

Thank you for your comments, it has been considered throughout the document.

Please be consistent in using abbreviations throughout the paper. I can see n-3, n-3FAs, n-3 FAs, n-3 FA; or C18:3 and C18:3 n-3.

Corrected

Please be consistent in using only American English or British English throughout the paper.

Revised and corrected. The article was revised by MDPI English editing service.

There are a few typos and spelling mistake.

The article was revised by MDPI English editing service.

Several sentences are unclear. They need to rewrite in a clear and concise way, or split them into different ones.

The article was revised by MDPI English editing service.

Some information in the manuscript need to provide at least a reference.

Reference included.

Some statements regarding the present study’s results are subjective and statistically incorrect.

Several cited references are not relevant. Please check again.

Revised

Specific comments:

Title:

I suggest replacing “Diets Supplemented” by “Dietary Lipid Source Supplementation”.

 Modified.

Affiliation:

Provide affiliations of I. Peñaranda, M.D. Garrido; and number the affiliations in the author list in an increasing order from the left to the right.

Modified.

A simple summary:

 Please provide a simple summary before the abstract section.

Included.

Abstract:

Lines 14-15: Please provide % of supplements.

This information has been added in the abstract.

“one including flaked linseed (FL) at 3.88% of dry matter, and the other salmon oil (SO) at 2.64% of dry matter for three periods of 21 d.”

Line 23: add “fatty” between “polyunsaturated” and “acids”.

Corrected.

The authors should a brief take-home message of the study at the end of the abstract.

Included

Introduction:

Lines 39-42: Need at least a reference.

Reference included.

Please provide some more information about linseed and salmon oil. Why did you choose them to become the study’s supplements?

We have improved the justification for the selection of supplements in the introduction:

“Furthermore, this work was designed as practical applied research in nutrition, so the omega-3-rich supplements selected for evaluation were commercially available as flaked linseed rich in ALA (Agrocava SL, Caravaca de la Cruz, Murcia, Spain), and a salmon fish oil product rich in DHA and EPA (Optomega-50, Optivite International Ltd., Nottinghamshire, United Kingdom).”

Line 61: Need at least 2 references.

 References included.

Materials and Methods:

Line 89: Why did the authors choose those different supplementing levels?

The percentages of incorporation of the supplements are different, since the lipid content differs between them (calcium soaps of palm oil FA, ingredient with a 84% of ether extract, flaked linseeds, seeds with a 34.7% of ether extract, and salmon oil, a product with 50% of ether extract), an aspect considered for the formulation of the diets. Now we have incorporated the ether extract content of these ingredients into the description of the materials and methods.

Lines 90-91: Which nutritional standards or system did you follow to state that all diets are the nutritional requirement? Please provide reference. What were the energy and protein density of the diets?

This paragraph has been completed and improved:

“All diets cover the nutritional requirements for lactating goats according the Fundación Española para el Desarrollo de la Nutrición Animal (FEDNA) [2010], and they were formulated to be iso-energetic and iso-nitrogenous.”

Lines 91-92: In table 1, the energy values of the diets were not found. Additionally, %CP were not similar among the treatments. As a result, the experimental diets seemed to be not isocaloric and isonitrogenous. Furthermore, the %EE are different among the diets. This may lead to unreliable results of the study. The authors may not be able to explain which factors actually affects their results. Please revise them and provide reasonable explanations.

In Table 1 the information of energy and protein content of the dietary diet is shown. In this table, the calculated and analyzed nutritive values of diets are displayed. For the Energy, the unit UFL (INRA system) was used, estimated according to data of ingredients reported by FEDNA. In this case, resulting of our formulation, diets were iso-energetic. In relation to Protein, the formulated diets resulted in values of Crude Protein not lower than 16.5%, PDIE and PDIN not lower than 10.3%, and PDIA not lower than 4.6%. When these diets were analyzed, the CP differences were lower than 1% (15.5%, 16.38%, and 16.17% for Control, FL and SO respectively). As consequence, Protein needs were cover by diets in a similar way. Therefore, for all the indicated in relation to CP, we can consider the diets as iso-nitrogenous. Regarding the EE, the diets were formulated to reach a 4.45% in DM (data that we have just included in the calculated values of the table). However, when the diets were analyzed, the EE value was 4.01%, 5.21%, and 4.18% for Control, FL and SO respectively; values close to the 4.5% calculated. Therefore, as this diet is a Total Mixed Ration, it is difficult to perform and homogenous analysis, although the results have shown a composition close to the calculated values.

Line 100-135: Although there are cited references, the sample collection and measurement methods need to have brief description. In the current version, they are too abstract.

The methods have been described in more detail.

Some equipment (e.g. a CoAguLiteTM optical sensor…) need to have the name of produced company, city and country. The information needs to be consistent in all of the equipment.

Revised and corrected.

Results:

Lines 206-208 and 211-214: Those statement are statistically incorrect. Please remove them.

Deleted

Please make a,b,c in the tables superscript.

Modified

Line 159: the unit of the clotting time in Table 3 is hour not Tmax.

Modified.

Discussion:

Lines 228-229: This is not correct. There was a tendency to reduce (p=0.1, Table 2), but not substantially reduced (p<0.05).

Modified

Lines 234-235: I could not find this information in the publication of Ferlay and Chillard (2020). Please check again.

The article specifies: We can also hypothesize that long-chain PUFAs from FO have a hypophagic effect, as suggested by Allen (2000). The authors have considered also citing in the manuscript the article that Ferlay and Chillard (2020) refers.

Line 254: Please remove: “less firm and”.

Removed.

Lines 257-258: This is a subjective statement. Please avoid.

Removed.

Line 264: Please provide a reference.

Reference included.

Lines 337-340: Please split this sentence into 2.

Revised and corrected.

Lines 342-343 and 348-351: They are not statistically correct.

Revised.

Conclusion:

This section has not been found in the current version. I suggest the author separate the last paragraph of the Discussion section and make it become the conclusion section.

Included.

Financial Support Statement:

Please provide the address of the financial sponsors.

Included.

Ethics approval:

I am concerned about the animal welfare in this experiment. Although the farm staff directly take care of the animals as normal, animal selection, the treatment assignments, changing in diets… might have effects on the animal welfare, which the farm staff could not professionally recognise and solve.

I enclose the certificate of the ethic committee signed that no assessment from this committee is necessary.

Authors Contributions:

Please follow the journal instruction, the author initials only.

Modified.

Reference list:

Please follow the author guideline of the journal.

Revised and modified. The article was revised by MDPI English editing service.

Round 2

Reviewer 2 Report

Comments and Suggestions for Authors

Dear authors.

Thanks for consider the suggestion made by the reviewers

The present work made by you is important and has on-farm implications for goat productive systems

Congrats

Author Response

The authors thanks the remarcable contribution of the reviewer.

Reviewer 4 Report

Comments and Suggestions for Authors

I am happy with the responses and revisions of the authors.

Please remove: "Several strategies can be used to modify the fatty acid profile of milk; the most common is lipid supplementation of the diet" in the Astract.

Check some typos, and format errors.

After correcting, for me the manuscript is accepted to publish.

Author Response

We deleted the sentence included in the abstract, and revise again the manuscript